# Nationwide COVID-19-EII Study: Incidence, Environmental Risk Factors and Long-Term Follow-Up of Patients with Inflammatory Bowel Disease and COVID-19 of the ENEIDA Registry

**DOI:** 10.3390/jcm11020421

**Published:** 2022-01-14

**Authors:** Yamile Zabana, Ignacio Marín-Jiménez, Iago Rodríguez-Lago, Isabel Vera, María Dolores Martín-Arranz, Iván Guerra, Javier P. Gisbert, Francisco Mesonero, Olga Benítez, Carlos Taxonera, Ángel Ponferrada-Díaz, Marta Piqueras, Alfredo J. Lucendo, Berta Caballol, Míriam Mañosa, Pilar Martínez-Montiel, Maia Bosca-Watts, Jordi Gordillo, Luis Bujanda, Noemí Manceñido, Teresa Martínez-Pérez, Alicia López, Cristina Rodríguez-Gutiérrez, Santiago García-López, Pablo Vega, Montserrat Rivero, Luigi Melcarne, Maria Calvo, Marisa Iborra, Manuel Barreiro de-Acosta, Beatriz Sicilia, Jesús Barrio, José Lázaro Pérez, David Busquets, Isabel Pérez-Martínez, Mercè Navarro-Llavat, Vicent Hernández, Federico Argüelles-Arias, Fernando Ramírez Esteso, Susana Meijide, Laura Ramos, Fernando Gomollón, Fernando Muñoz, Gerard Suris, Jone Ortiz de Zarate, José María Huguet, Jordina Llaó, Mariana Fe García-Sepulcre, Mónica Sierra, Miguel Durà, Sandra Estrecha, Ana Fuentes Coronel, Esther Hinojosa, Lorenzo Olivan, Eva Iglesias, Ana Gutiérrez, Pilar Varela, Núria Rull, Pau Gilabert, Alejandro Hernández-Camba, Alicia Brotons, Daniel Ginard, Eva Sesé, Daniel Carpio, Montserrat Aceituno, José Luis Cabriada, Yago González-Lama, Laura Jiménez, María Chaparro, Antonio López-San Román, Cristina Alba, Rocío Plaza-Santos, Raquel Mena, Sonsoles Tamarit-Sebastián, Elena Ricart, Margalida Calafat, Sonsoles Olivares, Pablo Navarro, Federico Bertoletti, Horacio Alonso-Galán, Ramón Pajares, Pablo Olcina, Pamela Manzano, Eugeni Domènech, Maria Esteve

**Affiliations:** 1Hospital Universitari Mútua Terrassa, 08221 Terrassa, Spain; olgabl27@gmail.com (O.B.); maceituno1@hotmail.com (M.A.); hermisende@gmail.com (P.M.); mariaesteve@mutuaterrassa.cat (M.E.); 2Centro de Investigación Biomédica en Red de Enfermedades Hepáticas y Digestivas (CIBEREHD), 28029 Madrid, Spain; javier.p.gisbert@gmail.com (J.P.G.); ajlucendo@hotmail.com (A.J.L.); caballol@clinic.cat (B.C.); mmanosa.germanstrias@gencat.cat (M.M.); luis.bujanda@osakidetza.net (L.B.); marisaiborra@hotmail.com (M.I.); fgomollon@gmail.com (F.G.); gutierrez_anacas@gva.es (A.G.); mariachs2005@gmail.com (M.C.); ericart@clinic.cat (E.R.); margalidasard.calafat@gmail.com (M.C.); eugenidomenech@gmail.com (E.D.); 3Hospital Gregorio Marañón, 218007 Madrid, Spain; drnachomarin@hotmail.com; 4Gastroenterology Department, Hospital Universitario de Galdakao, 48960 Galdakao, Spain; iago.r.lago@gmail.com (I.R.-L.); jcabriada@gmail.com (J.L.C.); 5Biocruces Bizkaia Health Research Institute, 48960 Galdakao, Spain; 6Hospital Universitario Puerta de Hierro, 28222 Majadahonda, Spain; isabel.veramendoza@gmail.com (I.V.); ygonzalezlama@gmail.com (Y.G.-L.); 7Hospital Universitario La Paz, 28046 Madrid, Spain; mmartina.hulp@salud.madrid.org; 8Hospital Universitario de Fuenlabrada, 28942 Fuenlabrada, Spain; ivangm79@gmail.com (I.G.); laujmarquez@gmail.com (L.J.); 9Instituto de Investigación Hospital Universitario La Paz (IdiPaz), 28046 Madrid, Spain; 10Department of Gastroenterology, Hospital Universitario de La Princesa, Universidad Autónoma de Madrid (UAM), 28049 Madrid, Spain; 11Instituto de Investigación Sanitaria Princesa (IIS-IP), 28006 Madrid, Spain; 12Hospital Universitario Ramón y Cajal, 28034 Madrid, Spain; pacomeso@hotmail.com (F.M.); mibuzon@gmail.com (A.L.-S.R.); 13Hospital Clínico San Carlos, 28040 Madrid, Spain; ctaxonera.hcsc@salud.madrid.org (C.T.); cristina.alba@telefonica.net (C.A.); 14Instituto de Investigación del Hospital Clínico San Carlos [IdISSC], 28040 Madrid, Spain; 15Hospital Universitario Infanta Leonor, 28031 Madrid, Spain; angelponmedicina@yahoo.es (Á.P.-D.); rocio_plaza@yahoo.es (R.P.-S.); 16Consorci Sanitari de Terrassa, 08227 Terrassa, Spain; MPiqueras@cst.cat (M.P.); RMena@cst.cat (R.M.); 17Hospital General de Tomelloso, 13700 Tomelloso, Spain; sonsolestamarit@hotmail.com; 18Hospital Clínic de Barcelona-IDIBAPS, 08036 Barcelona, Spain; 19Hospital Universitari Germans Trias i Pujol, 08916 Badalona, Spain; 20Fundación Hospital Universitario Doce de Octubre, 28041 Madrid, Spain; pilarmarmon123@telefonica.net (P.M.-M.); sonsolesolivares@hotmail.com (S.O.); 21Hospital Clinic Universitari de Valencia, 46010 Valencia, Spain; maiabosca@gmail.com (M.B.-W.); pnavarrocortes@gmail.com (P.N.); 22Hospital de la Santa Creu i Sant Pau, 08041 Barcelona, Spain; jgordillo@santpau.cat (J.G.); fbertoletti@santpau.cat (F.B.); 23Hospital Universitario Donostia, Instituto Biodonostia, 20014 San Sebastián, Spain; horacio.alonsogalan@osakidetza.eus; 24Universidad del País Vasco (UPV/EHU), 48940 Leioua, Spain; 25Hospital Universitario Infanta Sofía, 28703 San Sebastián de los Reyes, Spain; nmancenido@gmail.com (N.M.); rpajaresvi@gmail.com (R.P.); 26Hospital Virgen de la Luz, 16002 Cuenca, Spain; terechu.martinez@gmail.com (T.M.-P.); olcinapod@gmail.com (P.O.); 27Institut Hospital del Mar d’Investigacions Mèdiques (IMIM), Hospital del Mar, 08003 Barcelona, Spain; alicialg84@gmail.com; 28Complejo Hospitalario de Navarra, 31008 Pamplona, Spain; cristina.rodriguez.gutierrez@cfnavarra.es; 29Hospital Universitario Miguel Servet, 50009 Zaragoza, Spain; sgarcia.lopez@gmail.com; 30Complexo Hospitalario Universitario de Ourense, 32005 Ourense, Spain; pablo.vega.villaamil@sergas.es; 31Instituto de Investigación Sanitaria Valdecilla (IDIVAL), Hospital Universitario Marqués de Valdecilla, 39008 Santander, Spain; digrtm@humv.es; 32Hospital Universitari Parc Taulí, 08208 Sabadell, Spain; lmelcarne@tauli.cat; 33Hospital San Pedro-Logroño, 26006 Logroño, Spain; mciniguez@riojasalud.es; 34Hospital Universitario y Politécnico de la Fe de Valencia, 46026 Valencia, Spain; 35Hospital Clínico Universitario de Santiago, 15706 Santiago, Spain; manubarreiro@hotmail.com; 36Hospital Universitario de Burgos, 09006 Burgos, Spain; bsicilia4@gmail.com; 37Hospital Universitario Río Hortega (HURH), 47012 Valladolid, Spain; jbarrioa95@gmail.com; 38Hospital Universitario Fundación de Alcorcón, 28922 Alcorcón, Spain; jlperezc@fhalcorcon.es; 39Hospital Universitari de Girona Doctor Josep Trueta, 17007 Girona, Spain; dbusquets.girona.ics@gencat.cat; 40Instituto de Investigación Sanitaria del Principado de Asturias (ISPA), Hospital Universitario Central de Asturias, 33011 Oviedo, Spain; ipermar_79@hotmail.com; 41Hospital de Sant Joan Despí Moisès Broggi, 08970 Sant Joan Despí, Spain; merce.navarro@sanitatintegral.org; 42Hospital Álvaro Cunqueiro, 36213 Vigo, Spain; vicent.hernandez.ramirez@sergas.es; 43Hospital Universitario Virgen de la Macarena, Universidad de Sevilla, 41009 Sevilla, Spain; farguelles@telefonica.net; 44Hospital General Universitario de Ciudad Real, 13005 Ciudad Real, Spain; fernando.ramirez.est@gmail.com; 45Hospital Universitario de Cruces, 48903 Barakaldo, Spain; susana.meijidedelafuente@osakidetza.eus; 46Hospital Universitario de Canarias, 38320 San Cristobal de la Laguna, Spain; laura7ramos@gmail.com; 47Hospital Clínico Universitario “Lozano Blesa” and IIS Aragón, 50009 Zaragoza, Spain; 48Hospital Universitario de Salamanca, 37007 Salamanca, Spain; jmunozn@gmail.com; 49Hospital Universitari de Bellvitge, 08907 L’Hospitalet de Llobregat, Spain; tsuris@bellvitgehospital.cat; 50Hospital Universitario de Basurto, 48013 Bilbo, Spain; jone.ortizdezaratesagastagoitia@osakidetza.net; 51Consorcio Hospital General Universitario de Valencia, 46014 Valencia, Spain; josemahuguet@gmail.com; 52Althaia Xarxa Assistencial Universitària de Manresa, 08243 Manresa, Spain; jllao@althaia.cat; 53Hospital Universitario de Elche, 03203 Elche, Spain; marifegarciasepulcre@gmail.com; 54Complejo Asistencial Universitario de León, 24071 León, Spain; msierra.ausin@gmail.com; 55Hospital Clínico de Valladolid, 47003 Valladolid, Spain; mdura@saludcastillayleon.es; 56Hospital Universitario Álava, 01009 Gasteiz, Spain; sandra.estrecha@gmail.com; 57Hospital Virgen de la Concha, 49022 Zamora, Spain; amfcoronel@gmail.com; 58Hospital de Manises, 46940 Manises, Spain; hinova200@gmail.com; 59Hospital Universitario San Jorge, 22004 Huesca, Spain; lorenolivan@gmail.com; 60Instituto Maimónides de Investigación Biomédica de Córdoba (IMIBIC), Hospital Universitario Reina Sofía de Córdoba, 14004 Cordoba, Spain; evaiflores@gmail.com; 61Hospital General Universitario de Alicante, 03010 Alicante, Spain; 62Hospital Universitario de Cabueñes, 33394 Gijón, Spain; trastoy@hotmail.com; 63Hospital Universitario Son Llàtzer, 07198 Palma, Spain; nrull@hsll.es; 64Hospital de Viladecans, 08840 Viladecans, Spain; pgilabert.hv@gencat.cat; 65Hospital Universitario Nuestra Señora de Candelaria, 38010 Santa Cruz de Tenerife, Spain; dr.alejandrohc@gmail.com; 66Hospital Vega Baja de Orihuela, 03314 Alicante, Spain; aliciabbrotons@gmail.com; 67Hospital Universitario Son Espases, 07120 Palma, Spain; daniel.ginard@ssib.es; 68Hospital Universitari Arnau de Vilanova de Lleida, 25198 Lleida, Spain; eseseabi@gmail.com; 69Complexo Hospitalario de Pontevedra, 36071 Pontevedra, Spain; daniel.carpio.lopez@sergas.es

**Keywords:** COVID-19, SARS-CoV-2, inflammatory bowel disease

## Abstract

We aim to describe the incidence and source of contagion of COVID-19 in patients with IBD, as well as the risk factors for a severe course and long-term sequelae. This is a prospective observational study of IBD and COVID-19 included in the ENEIDA registry (53,682 from 73 centres) between March–July 2020 followed-up for 12 months. Results were compared with data of the general population (National Centre of Epidemiology and Catalonia). A total of 482 patients with COVID-19 were identified. Twenty-eight percent were infected in the work environment, and 48% were infected by intrafamilial transmission, despite having good adherence to lockdown. Thirty-five percent required hospitalization, 7.9% had severe COVID-19 and 3.7% died. Similar data were reported in the general population (hospitalisation 19.5%, ICU 2.1% and mortality 4.6%). Factors related to death and severe COVID-19 were being aged ≥ 60 years (OR 7.1, 95% CI: 1.8–27 and 4.5, 95% CI: 1.3–15.9), while having ≥2 comorbidities increased mortality (OR 3.9, 95% CI: 1.3–11.6). None of the drugs for IBD were related to severe COVID-19. Immunosuppression was definitively stopped in 1% of patients at 12 months. The prognosis of COVID-19 in IBD, even in immunosuppressed patients, is similar to that in the general population. Thus, there is no need for more strict protection measures in IBD.

## 1. Introduction

The COVID-19 pandemic hit Spain at the end of February 2020 and is far from under complete control. Data on affected cases and mortality are continuously updated [1]. There is evidence that patients suffering from inflammatory bowel disease (IBD) have a greater risk for infections, some of them opportunistic, mainly favoured by immunosuppressive treatment [2,3,4]. For that reason, experts on IBD, worried by the potential severity of COVID-19 in these patients, recommended, during the initial phases of the pandemic, that whenever possible, starting immunosuppressants should be delayed and treatment deescalated [5,6,7]. Notwithstanding this information, more than one year after the start of the pandemic, factors related to deleterious prognosis of COVID-19 in patients with IBD are essentially the same as those of the general population (mainly older age and comorbidities), whereas those on immunosuppressants do not appear to have a greater risk for severe COVID-19, except for corticosteroids [8]. In this sense, the international self-reported registry SECURE-IBD (https://covidibd.org (accessed on 24 August 2021)) has provided valuable clinical and therapeutic information [9]. Nevertheless, retrospective studies and registries have important limitations, such as reporting bias, over- or underrepresentation of the more severe cases of COVID-19, and the possibility of including confounding factors that may influence the results.

In addition, some studies have reported a low incidence of COVID-19 in patients with IBD [10], suggesting that IBD and the type of immunosuppressants administered for disease control do not represent risk factors for COVID-19. However, few of these studies are population based and do not address important environmental epidemiological risk factors, such as variability in the incidence of the infection in different regions within the same country. Neither do they address factors that may facilitate the infection, such as occupational risk, or those that may reduce the risk, as they may be specific isolation measures that could be recommended to a particular diseased population [10,11,12,13,14,15,16]. Likewise, the impact of COVID-19 on patients with IBD in the long term has not yet been explored.

The present study (COVID-19-EII study) was conducted in the setting of the ENEIDA project, the Spanish registry of patients with IBD, promoted by the Spanish Working Group on Crohn’s disease and ulcerative colitis (GETECCU) [17]. The aims of the present study were (1) to describe the incidence of COVID-19 in the ENEIDA registry, the geographical distribution of the infection compared with the distribution in the general Spanish population and exposure factors that may favour or prevent the infection (occupational risk and lockdown measures) during the first wave of the pandemic and (2) to describe the clinical characteristics and the disease course, including a 12-month follow-up after COVID-19.

## 2. Materials and Methods

### 2.1. Design

This was an observational prospective cohort study (COVID-19-EII) within the Spanish ENEIDA IBD registry. It included all patients with IBD who had COVID-19 between March and July 2020 (in the first wave) from the participant centres.

### 2.2. Study Population

The potential population was all patients with IBD registered in ENEIDA. Patients with COVID-19 were identified by an active search from their IBD unit (systematically addressing all the patients with IBD from the unit by email or phone call) or by direct notification from the patient itself, the family physician, the emergency department or the hospitalisation unit.

### 2.3. Data Collection

A prospective module hosted on the ENEIDA platform was specially designed for this study. Data collected included clinical baseline characteristics such as type of IBD, date of IBD diagnosis and Montreal’s classification [18], extraintestinal manifestations, family history of IBD and smoking behaviour at time of infection. The following comorbidities were specifically registered: cirrhosis, chronic renal failure, chronic obstructive pulmonary disease, heart disease, stroke, diabetes mellitus, arterial hypertension, dyslipidaemia, neoplasia, congestive heart failure, dementia, HIV, rheumatological disease or immune-mediated disease. Charlson’s index [19] was also calculated. We decided to explore both individual comorbidities and the Charlson comorbidity score, as these two approaches examine different aspects of comorbidity and are complementary. Variables measuring the exposure risk to SARS-CoV-2 included occupational risk (such as health care workers, teachers, basic services as supermarket cashiers, market clerks or pharmacy workers, police and firepersons, workers of closed institutions, veterinaries, animal control workers or conservation and forest technicians), compliance with lockdown measures, social distancing and the route of contagion. At the time of COVID-19 diagnosis, IBD activity was evaluated using the Harvey-Bradshaw index [20] or partial Mayo score [21]. The IBD therapeutic regimen was registered at the time of infection and up to 3 and 12 months before it: systemic steroid treatment, aminosalicylates, immunosuppressants (thiopurines, cyclosporine, methotrexate, tacrolimus and tofacitinib) and biologics (anti-TNF, vedolizumab and ustekinumab). Regarding COVID-19, the data collected included symptoms associated with the infection at the time of diagnosis, diagnostic procedures and specific treatment. The variables registered 3 and 12 months after COVID-19 were IBD activity and COVID-19 sequelae, both physical and psychological. To assess the impact of COVID-19 on IBD treatment, any change in medical therapy, including withdrawal of immunosuppression, both definitive and temporary, was collected during follow-up.

### 2.4. Definitions

COVID-19 diagnosis was based on a typical clinical picture consisting of fever (>38 °C), respiratory symptoms (cough and/or dyspnoea), anosmia or dysgeusia within the epidemiological setting. COVID-19 was considered confirmed by a positive diagnostic test including polymerase chain reaction (PCR) taken by nasopharyngeal swab or serology (IgM or IgG) for SARS-CoV-2. COVID-19 was considered probable in patients with a typical clinical picture but negative or lacking diagnostic tests. Asymptomatic patients with positive PCR or serology were not included.

It was considered that any patient had a good compliance with the lockdown measures when maintaining social distance by staying at home almost exclusively since 14 March 2020, the date the Spanish government ordered a total lockdown to prevent the spread of SARS-CoV-2.

Sequelae due to COVID-19 were any sign or symptom that the patient and/or physician considered related directly to COVID-19 and that was present at 3 and 12 months after infection.

### 2.5. Outcomes

To assess the disease course and clinical evolution of COVID-19, the following outcomes were registered and analysed: hospitalisation due to COVID-19, intensive care unit (ICU) admission, sequelae, severe COVID-19 and death. Severe COVID-19 was considered a composite variable that included ICU admission and/or use of active amines and/or respiratory distress and/or invasive oxygen therapy and/or death [22]. Cases with systemic inflammatory response syndrome were also registered. Data on outcomes of our study were compared to those registered on the SECURE-IBD registry (accessed on 24 August 2021) [9], considered the worldwide IBD registry on COVID-19. These outcomes were also compared to those of the general population taking into account data from Catalonia [23].

### 2.6. Ethical Considerations

The Scientific Committee of ENEIDA approved the study on 16 March 2020. It was also approved by the Ethics Committee of Hospital Universitari Mútua Terrassa (coordinating centre). Informed consent was obtained from all subjects. The patients were not identified by name in the publication, and no one, except the investigators of this study, had access to their local data, in accordance with the local Law of Personal Data Protection.

### 2.7. Statistical Analysis

Quantitative variables were compared with Student’s *t* test and the Mann–Whitney test, and the results are expressed as the means (±standard deviation) or median (±interquartile range (IQR) 25–75 percentiles). Quantitative variables were compared using Student’s *t* test for parametric data and the Mann–Whitney test for nonparametric data, while qualitative variables were compared using the Chi2 test or Fisher’s exact test, as appropriate. Univariate and multivariate logistic binary regression analyses were performed to explore the variables related to the need for hospitalisation, ICU admission, severe COVID-19, death and sequelae found at the 3- and 12-month follow-ups. The intensity of the significant associations was measured by calculating the OR and its 95% confidence interval. The multivariate models included significant variables in univariate analysis at the *p* < 0.1 level. In addition, for the outcomes with a small number of events (death and ICU admission), only the 2 most significant covariates in the univariate analysis were included. As the use of aminosalicylates was more frequent in patients with ulcerative colitis (UC) than in Crohn’s disease (CD), the model was adjusted to UC diagnosis when this drug was independently associated with a specific outcome.

Due to the great variability in the incidence of COVID-19 between the Spanish territories, the number of cases of both IBD and the general population are shown per province. Data on COVID-19 incidence in the general population as well as the age at COVID-19 diagnosis, age of hospitalized patients (including ICU admission) and age of patients with fatal outcomes were obtained from the National Centre of Epidemiology (CNE) [24,25]. The age- and sex-standardised incidence of every outcome in the IBD cohort was obtained based on all the patients actively followed in each participant centre of ENEIDA.

## 3. Results

A total of 73 out of the 86 centres adhered to the ENEIDA registry at the time of the study and decided to participate. This registry had, at that moment, 60,512 patients actively followed-up (data as for 15 July 2020), with 53,682 coming from the 73 participating centres (89% of the whole registry). Finally, 482 cases of COVID-19 were reported (251 males (52%); median 52 years (IQR: 42–61); cumulative incidence of 8.97 per 1000 patients with IBD, taking into account the population at risk from the participating centres). Ten centres that agreed to participate did not register any COVID-19 cases during the study period, despite being aware.

### 3.1. Clinical Baseline Characteristics

Table 1 and Table 2 show the most important clinical characteristics of the patients regarding IBD at the time of COVID-19 diagnosis. The activity of the disease and treatment are provided separately for UC and CD (Table 2). Notably, 80% of patients were in remission, and 9% had moderate–severe disease activity. Regarding IBD treatment at the time of the infection, 42% of the patients were on aminosalicylates, 5.4% were receiving systemic steroids, 36% were receiving immunosuppressants, 36% were receiving biologics and 12% were receiving combination therapy. Eleven percent of the patients required steroids within the 3 months before COVID-19 diagnosis.

Forty-four percent had at least one comorbidity, and 64% had a Charlson score of one or more. The most frequent comorbidity was arterial hypertension (22% (106/482)), followed by dyslipidaemia (15% (74/482)) and immune-mediated diseases (11% (53/482)) other than IBD (Appendix A).

### 3.2. Geografical Distribution of COVID-19 and Epidemiological Risk Factors of Exposure

The geographical distribution of cases is shown in Figure 1A, allowing a comparative approach with the incidence of COVID-19 in the Spanish general population (Figure 1B).

The majority of cases were reported in ENEIDA centres of the communities of Madrid and in those of the metropolitan area of Barcelona (red colour), where there is also the highest proportion of certified IBD units [26]. These two regions have the highest population density in Spain (844 and 743 inhabitants per km^2^, respectively, as of January 2021 [27]) and register the highest incidence of COVID-19 in the general population.

Compared to the Spanish population in March 2020 [24], the median age of IBD cases was similar: 52 years old (IQR, 42–61) in patients with IBD vs. 54 years old (IQR, 39–70) in the general population. When considering specific outcomes, a similar trend was observed: the age of hospitalised patients with IBD was 59 years old (IQR, 50–72) vs. 66 years old (IQR, 51–79) in the general population, and the age of ICU/death was 72 years old (IQR, 57–80) in patients with IBD vs. 70 years old (IQR, 59–80 years) in the general population

Regarding risk factors for COVID-19, almost half of the patients declared good adherence to lockdown measures (48% (229/482)). The circumstances that ensured an appropriate domiciliary lockdown were having preventive sick leave (31% (71/229)), being retired (26% (60/229)), doing telecommuting (18% (42/229)) and being unemployed (6.6% (15/229)). Table 3 summarizes the risk for SARS-CoV-2 infection related to occupational risk. 

Almost one-third of the patients had a job position considered as posing a high risk of infection, which was the main cause for not having proper adherence to lockdown measures. Health care professions were the most frequent occupational hazard (18% (85/482). Table 4 shows the relationship between infection and occupational risk in patients with and without good adherence to a total lockdown. 

Despite declaring good adherence, patients became infected mainly by intrafamilial transmission, particularly during the first 2 weeks (March 2020) after the Spanish government established lockdown.

### 3.3. COVID-19 Diagnosis and Treatment

Symptoms of COVID-19 diagnosis can be found in Appendix A, with fever (69% (336/482)) and cough (63% (305/482)) as the most frequently observed symptoms. Diarrhoea was reported in 26% (126/482) of patients, with no significant differences between patients with active or inactive IBD.

Appendix A includes the tests performed for COVID-19 diagnosis. Notably, 90% of IBD patients had a diagnostic test performed, whether PCR (80% (388/482)) or SARS-CoV-2 serology (35% (167/482)). Only 49 patients did not have any diagnostic test performed. It is also important to emphasize that 28% of patients (85/301) had a positive PCR after one or more negative PCRs, with no difference between immunosuppressed and non-immunosuppressed patients (19% vs. 21%, *p* = 0.67).

COVID-19 treatment followed the trend used and recommended by health authorities at that time. It included chloroquine/hydroxychloroquine in 41% (198/482), antibiotics in 38% (182/482), antivirals in 18% (88/482), systemic corticosteroids in 12% (58/482) and biologic therapy in 2.9% (14/482) (tocilizumab in 1.7% (8/482), interferon in 1.2% (6/482) and anakinra in 0.2% (1/482)). Antifungal therapy was used in 1.9% (9/482), vasoactive amines in 1.2% (6/482) and invasive oxygen therapy in 2.9% (14/482) of the patients (one patient received invasive oxygen therapy in a conventional floor due to occupation limitations of the ICU).

### 3.4. Outcomes

Patients attended their primary care facility in 55% of the cases (266/482), received emergency room assistance in 52% (251/482) and required hospitalisation due to COVID in 35% (167/482). Eleven patients (0.2%) required IBD-related hospitalisation during the study period. Twenty-four patients had respiratory distress (4.9%), 56 (12%) presented with systemic inflammatory response syndrome upon admission and 6.4% (31/482) presented with systemic inflammatory response syndrome during hospitalisation. Thirteen (2.5%) required ICU admission, 38 (7.9%) fulfilled the criteria of severe COVID-19 and 21 patients died during the study. Of those who died, 18 (3.7%) died due to COVID-19 and 3 due to causes other than COVID-19, 2 of them during the first wave of the pandemic (one case with signet-ring cell adenocarcinoma of digestive origin and one case of pulmonary neoplasia) and the third 9 months after COVID-19 infection due to urinary sepsis. Only one death occurred outside the hospital (80-year-old female with inactive ileal CD treated with oral aminosalicylates). Compared to the general population in Spain, the severe outcomes were similar, with a mortality proportion of 4.6% and ICU requirement proportion of 2.1% with a slightly lower proportion of patients requiring hospitalisation (19.5%) [23].

Univariate and multivariate analyses of factors associated with hospitalisation, ICU admission, severe COVID-19 and death are detailed in Appendix A. In that case, predictive factors for hospitalisation due to COVID-19 were being 50 years of age or more (OR 2.09, 95% CI 1.3–3.4), having at least one comorbidity (OR 2.28, 95% CI 1.4–3.6) and being treated with steroids for IBD within the 3 months before COVID-19 diagnosis (OR 1.3, 95% CI 1.1–1.6). Predictors for ICU admission were having a Charlson score of at least 2 (OR 5.4, 95% CI 1.5–20.1) and the use of aminosalicylates (OR 4.6, 95% CI 1.2–17). However, when adjusted for diagnosis, the effect of aminosalicylates disappeared (OR 3.6, 95% CI 0.85–15.2, *p* = 0.08). Independent risk factors related to death due to COVID-19 were being 60 years of age or more (OR 7.1, 95% CI 1.8–27.4) and having at least 2 comorbidities (OR 3.9, 95% CI 1.3–11.6). The only predictor for severe COVID-19 was being 60 years of age or more (OR 4.59, 95% CI 1.3–15.9), while having CD with an inflammatory behaviour was protective for this outcome (OR 0.29, 95% CI 0.09–0.89). There were no differences in the proportion of hospitalisation, ICU admission, severe COVID-19 or death between patients found under active search (18% of the centres) vs. those that were found by direct notification from the patient itself, the family physician, the emergency department or the hospitalisation unit (data not shown). The proportion of patients under specific IBD treatment, taking into account each predefined outcome, is shown in Table 5.

There were no differences in any outcomes between patients with probable or confirmed COVID-19 (data not shown). To compare the results of the present study with those of the SECURE-IBD (accessed on 24 August 2021) [9], the results of the two cohorts are provided in the same table. In Appendix A, we show the incidence of adverse outcomes of COVID-19 in patients with IBD, taking into account sex and age. Of note, being ≥50 years old increases the incidence of the reported adverse outcomes from 3 to 7, with a greater frequency in males than in females. 

### 3.5. Follow-Up

IBD treatment and the presence of sequelae related to COVID-19 at the 3- and 12-month follow-up are described in Table 6. 

Only 13% of patients had short-term immunosuppression withdrawal due to COVID-19 (12% partial and 1.4% definitive), and only 1% kept their withdrawal of immunosuppression in the long term.

At the 3-month follow-up, 65 patients (13%) presented COVID-19 sequelae, of which 4% (20/482) were psychological and 11% (55/482) were physical (Table 6). At the 12-month follow-up, 72 patients (15%) were considered to have sequelae, of which 3.1% (15/482) were psychological and 14% (67/482) were physical. The most frequent physical sequelae were asthenia, myalgia/arthralgia and anosmia. The only predictive factor for having physical sequelae at the 3-month follow-up was the use of steroid treatment for IBD within the 3 months before COVID-19 (OR 1.4, 95% CI 1.07–1.7) (Appendix A). No predictive factor for physical sequelae was found at the long-term follow-up (12 months) (Appendix A). Only one patient reported SARS-CoV-2 reinfection 6 months after the index diagnosis.

IBD activity both at COVID-19 diagnosis and at 3- and 12-month follow-ups is shown in Figure 2. Two-thirds of patients (68%, 330/482) remained in remission throughout the study period.

## 4. Discussion

We report the largest cohort of patients with IBD and symptomatic COVID-19 prospectively recruited with a one-year follow-up after infection. This is a national, multicentre study that was conducted within the ENEIDA project and included 482 patients with COVID-19 among 53,682 patients with IBD, giving a cumulative incidence of 8.97 per 1000. These data are in the upper limit of the wide range of incidence previously described [10,14,28], ranging from 0.95 [29] to 100 per 1000 [15]. The universal access to health care within the National Health System in Spain (less than 3% of IBD patients with private insurance never use public services [30]) and the adherence of most centres to the nationwide certification programme in IBD [26] provide homogeneity to the cohort, minimizing potential bias in the clinical characteristics. We did not calculate the comparative incidence of COVID-19 between the IBD cohort and data obtained from the general population because the identification of cases did not use the same methodology. However, it is clearly observed in Figure 1 that the number of cases with COVID-19 and IBD is the highest in areas with the highest incidence of infection. The high variability in incidence between relatively close geographical areas was also observed, with a maximum in those areas with the highest population density. We decided to exclude asymptomatic patients, as there was no policy for a universal testing in our patients nor in the general population during the first wave. Therefore data on asymptomatic patients with IBD and positive SARS-CoV-2 test are scarce and can imply a selection bias.

It has been suggested that patients with IBD are at lower risk of having COVID-19, and most studies support this assertion [10,31,32,33,34,35]. However, only a few of these studies performed in Denmark and Sweden were population based [36,37], which is how differences in the incidence of infection between population groups should be addressed. In our cohort, we recorded the influence of occupational risk, lockdown strategies and other risk factors, in addition to IBD treatment, that may have influenced, either increasing or decreasing, the risk of contagion. We observed, for example, that one-third of infected patients received special protection measures such as sick leave simply because they were considered a risk group. However, many patients became infected, perhaps because they were infected early before mandatory lockdown or through close family-infected contact. Lockdown has been demonstrated to be the most effective strategy precluding SARS-CoV-2 expansion [38,39,40], also in patients with IBD [41]. Thus, the low incidence of COVID-19 in IBD cohorts does not necessarily reflect a lower susceptibility to infection but a higher protection attitude towards IBD patients based on recommendations [6,42] or because they spontaneously adopt more rigorous self-protecting measures [43]. It has been shown that patients under biologic drugs perceive themselves to be at greater risk of SARS-CoV-2 infection, but, despite that, they do not perform more strict social distancing practices than patients with IBD without biologics or in remission [43]. However, in our cohort, only 49% exhibited good adherence to lockdown measures. Currently, data on the exposure environment of noninfected patients are very limited [44]. The study also showed that one-third of IBD patients with COVID-19 had occupational risks, mainly working in health care facilities. It has been reported that health care providers bore a great burden during the pandemic, as shown in data coming from Italy and China [22,45]. Nonetheless, we did not find that occupational risk or failure to meet lockdown measures were predictors for a worse evolution of COVID-19 in patients with IBD.

The reported outcomes of COVID-19 are highly variable [10,28,33,36,46] and may depend on many factors, such as age, comorbidity, ethnicity, socioeconomic status and the quality of health care. In our cohort, as in others, the most important factors influencing outcomes were age and comorbidity. Thirty-five percent of the patients required hospitalisation, 7.9% had severe COVID-19, 2.5% required ICU admission and 3.7% died. These figures are similar to those extracted from the general population in Spain [23] (hospitalisation 19.5%, ICU 2.1% and mortality 4.6%) and are also identical to those reported in the first publication of SECURE-IBD, including the first 525 registered cases [8].

Age is known to be one of the most consistent risk factors for severe COVID-19 and death worldwide [47,48], both in the general population and in patients with IBD [10,49]. We found that patients 50 years old or older were at greater risk for hospitalization, while those 60 years old or older were more prone to severe COVID-19 or death due to COVID-19. Comorbidities have also been considered the other important risk factor for deleterious COVID-19 evolution, both in patients with IBD [16,29,33,36,46] and in the general population [22,50], and we also found consistent results in our cohort. Thus, differences in the percentages of parameters of the severity of COVID-19 between cohorts can be largely explained by these two factors, not only by themselves. The median age of patients in the initial SECURE-IBD cohort was 41 years versus 52 years in our cohort, and the percentage of comorbidities was 36% versus 44%, respectively. Data from SECURE-IBD accessed on 24 August 2021 [9], showed significantly improved outcomes compared to previous outcomes, with a fifty percent reduction in indicators of severity, including hospitalisation, severe COVID-19 and death (more than 70% of the patients were younger than 49 years). Although it can be speculated that improved knowledge of COVID-19 management may account for a better outcome over time, important selection bias accounting for differences in this type of registry cannot be ruled out.

The only protective factor for severe COVID-19 was CD with inflammatory behaviour. As this was found to be independent of IBD activity and treatment, we speculate that this is because this pattern has less disease burden, as it was also related to a shorter duration of the disease (11 years of disease in inflammatory IBD vs. 18 years in stricturing or penetrating disease, *p* < 0.0001).

As stated previously, IBD-related immunosuppression has not been found to be a risk factor for severe COVID-19 or death [28,29,51]. Our cohort confirms that there is no relationship between anti-TNF or any other form of therapeutic immunosuppression and COVID-19 severity. Some authors suggested that the risk of hospitalisation is higher in patients under biologics, but this may reflect a precaution more than COVID-19 severity itself, as overall mortality related to COVID has not been demonstrated to be increased in ours and other previous studies [28,36,46,51,52]. This consistent evidence reinforces the message that biologics can be safely continued in most cases. The use of steroids in this pandemic has been controversial [53,54]. In contrast to SECURE-IBD [8], we did not find that current treatment with systemic steroids was related to a worse COVID-19 evolution, an outcome that we have previously found related to other relevant infections in patients with IBD [3]. This might be due to the small proportion of patients under this treatment (5.4%) or the type of schedule administered. However, the use of steroids three months before COVID-19 diagnosis was an independent risk factor for hospitalisation and physical sequelae (at the 3-month follow-up). This could be indirectly related to a probable active IBD that was challenging to treat and required the use of systemic steroids.

Finally, we describe the evolution of patients at 3 and 12 months after COVID-19. Thirteen percent withdrew IBD medication during COVID-19. This is less than previously reported, ranging between 27% [15] and 34% [55]. This is certainly encouraging, as the first guidelines on the treatment of patients with IBD during the pandemic were very clear in recommending the withdrawal of immunosuppressants and biologics in infected individuals [6]. However, as our experience increased, it was assured that the inappropriate cessation of effective agents for IBD treatment due to unjustified fear of adverse events could lead to IBD relapse and then to the use of steroids or hospitalisation, thus increasing the risk of COVID-19 exposure and infection. The last outcome that we explored was COVID-19 sequelae, present in 13% at 3 months and 15% at 12 months in our cohort. The only predictive factor for having physical sequelae at the 3-month follow-up was the use of steroid treatment for IBD 3 months before COVID-19 (OR 1.4, 95% CI 1.07–1.7). It has been shown that severe sequelae were lower in patients with IBD when matched to non-IBD controls [49], so IBD does not seem to be a disease linked to more sequelae due to COVID-19. In contrast, a recent population-based Danish study has suggested that sequelae are a common phenomenon, affecting almost 44% of patients [56]. It has to be noted that this study accessed sequelae only in 222 of the 516 patients included and that they were self-reported sequalae, limiting the confirmation of a real effect that might be caused by COVID-19. In addition, another study found that patients with IBD, hospitalized due to COVID-19, have a greater risk of severe infections requiring further hospitalisation [57], a situation that has not been found in our cohort.

Our study has several limitations to be aware of. First, only 18% of centres performed an active search of cases (82% of cases were collected by direct notification from the patient itself, the family physician, the emergency department or the hospitalisation unit, as stated in the Methods section). Thus, it is possible that very mild cases did not consult their IBD unit; thus, mild COVID-19 cases might be underrepresented. However, data on hospitalisation, severe COVID-19, ICU admission and death from the general population are similar to our cohort and we did not find differences in specific outcomes between these two searching strategies; therefore, this bias is probably small. On the other hand, the majority of participant centres were certified IBD units that had open access to their outpatient clinics, nurse-led advice lines and/or emails for emergencies as mandatory quality criteria [26]. Second, this study, including only the first wave of an ongoing pandemic, is observational and cannot establish causation or account for unmeasured confounders. Finally, discovering the real number of infected patients remains a global challenge because PCR or serology were not performed universally at the beginning and there is evidence of false negative results. This occurred in 10% of this cohort. Notwithstanding these issues, there are some important strengths. First, it has national coverage with active participation of almost 90% of the IBD units from the ENEIDA registry. However, the main strength relies on prospective data, collecting IBD activity, and other important variables. In addition, although nationwide series on COVID-19 and IBD have been published before [29,33], our study is the largest cohort of patients with IBD and COVID-19 with the longest follow-up after infection.

## 5. Conclusions

In conclusion, we have shown that occupational risk and intrafamilial transmission are relevant epidemiological risk factors and that a high proportion of patients receive preventive sick leave. We have also demonstrated that IBD does not worsen COVID-19 prognosis, even when immunosuppressants and biological drugs are used. Age and comorbidity are the most important prognostic factors for more severe COVID-19 in patients with IBD and are even more relevant than epidemiological risk factors such as occupational risk. Finally, COVID-19 is not a condition that affects the prognosis of IBD or its treatment, either in the short or the long term and is not a cause of significant sequelae in patients already suffering from IBD. Therefore, there is no need for more strict protection measures than those adopted for the general population.

## Figures and Tables

**Figure 1 jcm-11-00421-f001:**
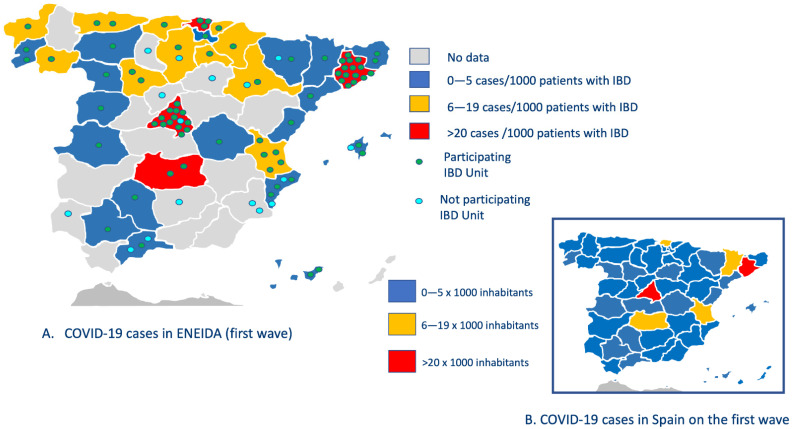
Geographic distribution of COVID-19 cases in the ENEIDA registry and comparison with the general Spanish population in the first wave of the pandemic.

**Figure 2 jcm-11-00421-f002:**
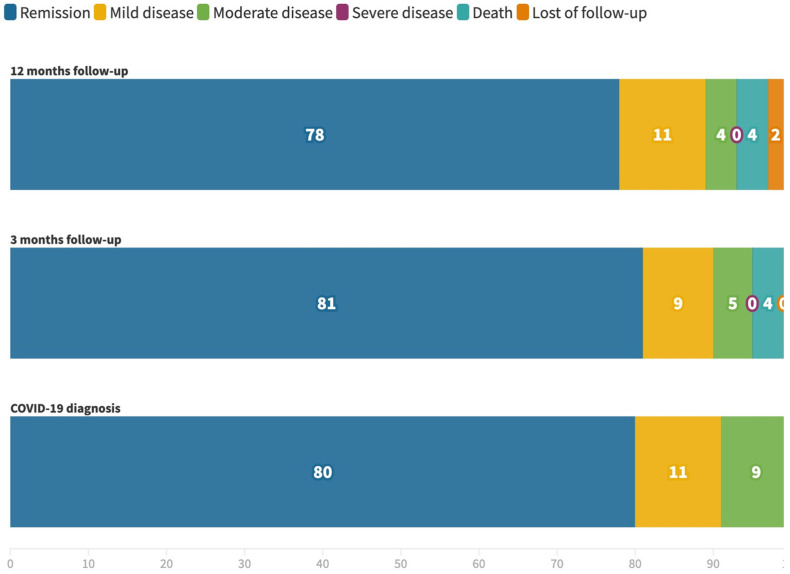
Inflammatory bowel disease activity at COVID-19 diagnosis and 3- and 12-month follow-up.

**Table 1 jcm-11-00421-t001:** Clinical baseline characteristics regarding inflammatory bowel disease with COVID-19.

Clinical Characteristics	Cases *n* = 482
Gender	
Male	251 (52)
Female	231 (48)
Age at COVID-19 diagnosis	52 years (IQR 42–61)
IBD duration at COVID-19 diagnosis	12 years (IQR 6–19)
Type of IBD, *n* (%)	
Crohn’s disease	247 (51)
Ulcerative colitis	221 (46)
Unclassified colitis	14 (2.9)
Ulcerative colitis extent (%)	
E1	43 (19)
E2	80 (36)
E3	98 (44)
Crohn’s disease location, *n* (%)	
L1	114 (46)
L2	43 (17)
L3	88 (36)
L4 (isolated)	3 (1.2)
Crohn’s disease behaviour, *n* (%)	
B1	144 (58)
B2	71 (29)
B3	47 (19)
Perianal	59 (24)
B1 + perianal	29 (12)
B2 + perianal	18 (7.3)
B3 + perianal	20 (8.1)
Extraintestinal manifestation, *n* (%)	125 (26)
Family history of IBD, *n* (%)	64 (13)
Smoking behaviour, *n* (%)	
Active	53 (11)
Former smoker	137 (28)
Never smoker	268 (56)

IQR: interquartile rate, IBD: inflammatory bowel disease, L1: ileal, L2: colonic, L3: ileocolonic, L4: upper gastrointestinal tract; B1: inflammatory behaviour, B2: stricturing behaviour, B3: penetrating behaviour.

**Table 2 jcm-11-00421-t002:** Inflammatory bowel disease activity and treatment at time of COVID-19 diagnosis.

	IBD (Total) *n* = 482	Crohn’s Disease *n* = 247	Ulcerative Colitis *n* = 221	*p*-Value *
IBD Activity at COVID-19 Diagnosis				
Clinical remission	385 (80)	200 (81)	173 (78)	0.35
Active disease	97 (20)	47 (19)	48 (22)	0.35
Mild	53 (11)	26 (10.5)	26 (12)	
Moderate	42 (8.7)	21 (8.5)	20 (9)	
Severe	2 (0.4)	0	2 (0.9)	
IBD treatment				
None, *n* (%)	62 (13)	37 (15)	23 (10.4)	0.15
5-aminosalicylates, *n* (%)	202 (42)	49 (20)	143 (65)	<0.0001
Oral (oral and topic)	197 (41)	49 (20)	138 (62)	
Topical (exclusive)	5 (1)	0	5 (2.3)	
Monotherapy	131 (27)	31 (12)	91 (41)	
Systemic steroids 3 months before COVID-19 (oral or intravenous), *n* (%)	53 (11)	30 (12)	21 (9.5)	0.36
Systemic steroids, *n* (%)	26 (5.4)	16 (6.4)	8 (3.6)	0.37
Immunosuppressants (in monotherapy), *n* (%)	113 (23)	65 (26)	56 (25)	0.03
Azathioprine	90 (19)	54 (22)	46 (21)	0.04
Mercaptopurine	8 (1.7)	4 (1.6)	2 (0.9)	0.16
Cyclosporine	1 (0.2)	0	1 (0.4)	0.96
Methotrexate	9 (1.9)	6 (2.4)	3 (1.3)	0.06
Tacrolimus	1 (0.2)	1 (0.4)	0	1
Tofacitinib	4 (0.8)	0	4 (1.8)	0.10
Biologics (in monotherapy), *n* (%)	117 (22)	72 (29)	35 (16)	0.04
Anti-TNF	71 (15)	42 (17)	19 (8.6)	<0.0001
Vedolizumab	25 (5.2)	12 (4.8)	13 (5.9)	0.75
Ustekinumab	21 (4.3)	18 (7.3)	3 (1.3)	0.001
Combotherapy, *n* (%)	59 (12)	45 (18)	14 (6.3)	0.02
Anti-TNF plus thiopurines	37 (7.7)	28 (11)	9 (4.1)	0.02
Anti-TNF plus methotrexate	9 (1.9)	6 (2.4)	3 (1.3)	0.62
Vedolizumab plus thiopurines	5 (1)	4 (1.6)	1 (0.4)	0.73
Vedolizumab plus methotrexate	1 (0.2)	0	1 (0.4)	0.78
Ustekinumab plus thiopurines	5 (1)	5 (2)	0	0.55
Ustekinumab plus methotrexate	2 (0.4)	2 (0.8)	0	0.98

* Comparison between Crohn’s disease and ulcerative colitis. IBD: inflammatory bowel disease; TNF: tumour necrosis factor.

**Table 3 jcm-11-00421-t003:** Epidemiological factors of SARS-CoV-2 infection and occupational risk.

**Route of Contagion, *n* (%)**	
Unknown	242 (50)
Intrafamilial transmission	108 (22)
Occupational	96 (20)
Travel	8 (1.7)
**Occupational Risk, *n* (%)**	133 (28)
Healthcare	85 (18)
Basic services (supermarket cashiers, market clerks, pharmacy)	18 (3.7)
Education	15 (3)
Police and fireperson	5 (1)
Closed institutions	2 (0.4)
Veterinary, animal control worker or conservation and forest technician	4 (0.8)

**Table 4 jcm-11-00421-t004:** Relationship between the route of contagion and occupational risk in patients with and without a total lockdown.

Risk Variable	Patients with Total Lockdown (*n* = 229)	Patients without Total Lockdown (*n* = 225)	*p*-Value
Route of contagion, *n* (%)			
Intrafamilial transmission	70 (31)	38 (17)	0.001
Infection on March 2020	47 (20)	26 (12)	0.007
Infection on April–July 2020	23 (10)	12 (5.3)	0.034
Occupational	19 (8.3)	77 (34)	<0.0001
Infection on March 2020	19 (8.2)	48 (21)	<0.0001
Infection on April–July 2020	0	29 (13)	<0.0001
Travel	3 (1.3)	5 (2.2)	0.69
Infection on March 2020	3 (1.3)	5 (2.2)	0.69
Infection on April–July 2020	-	-	
Unknown	137 (60)	105 (47)	0.005
Infection on March 2020	83 (36)	69 (31)	0.167
Infection on April–July 2020	54 (24)	36 (16)	0.003
Occupational risk, *n* (%)			
Occupational risk (all)	38 (17)	92 (41)	<0.0001
Healthcare	18 (7.9)	65 (29)	<0.0001

**Table 5 jcm-11-00421-t005:** Outcome of the COVID-19-EII cohort and treatment administered. The results are compared with those reported in the SECURE-IBD cohort (as for 24 August 2021) [9] ^¶^.

A. Outcomes										
**Outcome**			**Hospitalised**		**ICU Admission**		**Severe COVID-19**		**Death**	
			**COVID-19-EII** ***n* = 482**	**SECURE-IBD ** ***n* = 6438**	**COVID-19-EII** ***n* = 482**	**SECURE-IBD** ***n* = 6438**	**COVID-19-EII** ***n* = 482**	**SECURE-IBD** ***n* = 6438**	**COVID-19-EII** ***n* = 482**	**SECURE-IBD** ***n* = 6438**
			*n* = 168 (35%)	*n* = 977 (15%)	*n* = 13 (2.6%)	*n* = 184 (2.8%)	*n* = 38 * (7.8%)	*n* = 257 ** (3.9%)	*n* = 18 (3.7%)	*n* = 104 (1.6%)
B. Treatment taking into account each specific outcome										
**DRUG**	**Total**		**OUTCOMES**							
			**Hospitalised**		**ICU**		**Severe COVID-19**		**Death**	
**Cohort**	**COVID-19-EII** ***n* = 482**	**SECURE-IBD** ***n* = 6438**	**COVID-19-EII** ***n* = 168**	**SECURE-IBD** ***n* = 977**	**COVID-19-EII** ***n* = 13**	**SECURE-IBD** ***n* = 184**	**COVID-19-EII *** ***n* = 38**	**SECURE-IBD **** ***n* = 257**	**COVID-19-EII** ***n* = 18**	**SECURE-IBD** ***n* = 104**
5-aminosalicylates,	202 (42)	1924 (30)	79 (47)	411 (42)	10 (77)	81 (44)	24 (63)	118 (46)	9 (50)	53 (51)
Alone	131 (27)	-	52 (31)	-	5 (38)	-	16 (42)	-	6 (33)	-
With other IBD drugs	71 (15)	-	27 (16)	-	5 (28)	-	8 (21)	-	3 (17)	-
Systemic steroids	26 (5.4)	414 (6.4)	11 (6.5)	146 (15)	1 (7.7)	41 (22)	4 (10.5)	53 (21)	1 (5.6)	28 (27)
Thiopurines (monotherapy)	108 (22)	551 (8.6)	38 (23)	114 (12)	3 (23)	26 (14)	5 (13)	33 (13)	1 (5.6)	11 (10.6)
Methotrexate (monotherapy)	9 (1.9)	49 (0.8)	4 (2.4)	13 (1.3)	1 (7.7)	1 (0.5)	2 (5.3)	3 (1.2)	1 (5.6)	2 (1.9)
Anti-TNF (monotherapy)	71 (15)	2082 (32)	11 (6.5)	178 (18)	2 (15)	24 (13)	3 (7.9)	31 (12)	2 (11)	10 (9.6)
Anti-TNF in combotherapy	46 (9.5)	636 (9.9)	20 (12)	91 (9.3)	0	17 (9)	2 (5.3)	21 (8.2)	2 (11)	6 (5.8)
Vedolizumab	30 (6.2)	706 (11)	15 (8.9)	94 (9.6)	0	21 (11)	4 (10.5)	28 (10.9)	2 (11)	9 (8.6)
Ustekinumab	28 (5.8)	602 (9)	6 (3.6)	50 (5.1)	1 (7.7)	9 (4.9)	1 (2.6)	11 (4.3)	1 (5.6)	5 (4.8)
Tofacitinib	4 (0.8)	103 (1.6)	2 (1.2)	12 (1.2)	0	4 (2.2)	0	4 (1.5)	0	1 (0.9)

ICU: intensive care unit, TNF: tumour necrosis factor. ^¶^ Brenner EJ, Ungaro RC, Colombel JF, Kappelman MD. SECURE-IBD Database Public Data, https://covidibd.org/current-data/ accessed on 24 August 2021. * Severe COVID-19 for the COVID-19-EII study: composite of intensive care unit admission and/or use of active amines and/or respiratory distress and/or invasive oxygen therapy and/or death. ** Severe COVID-19 for the SECURE-IBD registry: composite of intensive care unit admission and/or use of ventilator and/or death.

**Table 6 jcm-11-00421-t006:** Sequelae at 3- and 12-months of COVID-19 and the impact of the infection on changes in therapeutic regimens for IBD.

	3 Months Follow-Up (*n* = 462)	12 Months Follow-Up (*n* = 451)
COVID-19 sequelae, *n* (%)	65 (14)	72 (15)
Psychologic sequelae, *n* (%)	20 (4.3)	15 (3.3)
Physical sequelae, *n* (%)	55 (12)	67 (15)
Asthenia	21 (4.5)	22 (4.8)
Myalgia/Arthralgia	7 (1.5)	3 (0.7)
Anosmia	4 (0.9)	7 (1.5)
Dyspnoea	4 (0.9)	6 (1.3)
Odynophagia	2 (0.4)	2 (0.4)
Dysgeusia	2 (0.4)	2 (0.4)
Hair loss	1 (0.2)	1 (0.2)
Pulmonary fibrosis	1 (0.2)	3 (0.6)
Bronchial hyperreactivity	1 (0.2)	1 (0.2)
Deep venous thrombosis/pulmonary thrombosis	1 (0.2)	1 (0.2)
Headache	1 (0.2)	6 (1.3)
Obstructive pulmonary disease	1 (0.2)	3 (0.6)
Paraesthesia	1 (0.2)	3 (0.6)
Wegener’s vasculitis	-	1 (0.2)
Immunosuppression withdrawal, *n* (%)	65 (14)	6 (1.3)
Transient	58 (13)	1 (0.2)
Definitive	7 (1.5)	5 (1.1)
De-escalation from combo to monotherapy	13 (2.9)	1 (0.2)
Patients requiring Immunosuppression initiation or modification, *n* (%)	12 (2.6)	43 * (9.5)
Systemic corticosteroids	1 (0.2)	7 (1.6)
Thiopurines	2 (0.4)	5 (1.1)
Methotrexate	0	1 (0.2)
Anti-TNF	5 (1)	18 (4)
Vedolizumab	1 (0.2)	7 (1.6)
Ustekinumab	2 (0.4)	12 (2.7)
Tofacitinib	1 (0.2)	6 (1.3)

* Some patients required more than one new immunosuppressant (combined or sequential); therefore, this number expresses the total number of patients that required immunosuppression initiation/modification from 3 to 12 months after COVID-19. IBD = inflammatory bowel disease; TNF: tumour necrosis factor.

## Data Availability

The data underlying this article are available in the article and Appendix A.

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
