# Peer review of "Nationwide COVID-19-EII Study: Incidence, Environmental Risk Factors and Long-Term Follow-Up of Patients with Inflammatory Bowel Disease and COVID-19 of the ENEIDA Registry"

_jcm, 2022, doi:10.3390/jcm11020421_

Round 1

Reviewer 1 Report

Thank you for the opportunity to peer review your manuscript.

Note I am more of an IBD expert than Covid-19 expert so you have the right of reply to what I query regarding Covid-19.

I have some queries or suggestions below.

Abstract:

I do not think you need to have 1), 2), 3), etc. Use the appropriate subheadings as outlined by the journal.

One of your conclusions in the abstract is "the prognosis... is similar to that in the general population." You may want to mention in the methods of the abstract how you compared to the general population.

You seem to report 3.7% of cases died; this seems very high as the IFR is not that generally. What is the IFR in Spain and how does your reported IFR of 3.7% differ? Could your death rate be inflated by sample biases? 

I assume these figures precede vaccination.

English/grammar needs to be checked in a few places. e.g., line 130 "whereas [those on] immunosupressants do not appear"

139-140, "few of these studies are not population based"... please check this sentence; I suspect you are trying to say "few of these studies are population based..."

line 162-163  by [an] active search

I want to make sure I understand line 200-201. You are saying if someone had no symptoms but tested postive they were excluded? Why is this? Is this a World Health Organisation recommendation? Aren't many covid cases asymptomatic? Conversely, why did you include symptomatic negative cases but not asymptomatic postive cases? This could be inflating your death rate of 3.7% could it not? Do you know how many tested positive but were asymptomatic? If so, what happens to the analyses if you include those people? Please help the reader understand this better.

At line 246, what do you mean that there are "60,512 patients actively controlled"? Does that mean each of the 60,512 has a match from the general population?

SECURE-IBD, whilst mentioned in the introduction, somewhat appears out of nowhere in Table 5. You may want to mention in your methods that you do some comparing to SECURE-IBD and describe it briefly (i.e. a database of people with IBD and Covid-19).

You do acknowledge some of your sampling bias issues at line 142. I have noted I wondered why you excluded those who tested positive but had no symptoms at line 200 methods. You also mention only 18% of centres performed an active search of cases. Did you think to compare those who were found via an active search and those who were found via the other means mentioned at line 143, as from that you will be able to see if data collection methods skewed any of your results somehow. e.g., maybe those actively found were less severe.

Author Response

Reviewer 1.

Thank you for the opportunity to peer review your manuscript.

Note I am more of an IBD expert than Covid-19 expert, so you have the right of reply to what I query regarding Covid-19.

I have some queries or suggestions below.

Abstract:

  1. I do not think you need to have 1), 2), 3), etc. Use the appropriate subheadings as outlined by the journal.

We have removed the numbers. However, the submission instructions state to write a single paragraph without headings. That is the reason why we did not use any heading or subheading, as suggested by the reviewer.

  1. One of your conclusions in the abstract is "the prognosis... is similar to that in the general population." You may want to mention in the methods of the abstract how you compared to the general population.

We have added the following phrase in the abstract: “Results were compared with data of the general population from the National Centre of Epidemiology”. However, due to the lack of space we had to reduce the abstract (removing information on the cumulative incidence, for example) to fit the requirements.

  1. You seem to report 3.7% of cases died; this seems very high as the IFR is not that generally. What is the IFR in Spain and how does your reported IFR of 3.7% differ? Could your death rate be inflated by sample biases? 

As it is stated in the Discussion section (page 21, lines 51-53), this data is comparable to the general population in Spain. In reference 49 (clinical characteristics of hospitalised COVID-19 patients in Spain) that included more than 15,000 patients, death was found in 4.7% of patients between 50 and 59 years. The median age of our cohort is 52 years.

  1. I assume these figures precede vaccination.

Exactly. As stated in the Methods section (page 4, lines 164-166), this research considers the first wave of the pandemic in Spain: from March to June/2020. Vaccination started on December 27th 2020.

  1. English/grammar needs to be checked in a few places. e.g., line 130 "whereas [those on] immunosupressants do not appear"

Done, thank you. Now it is in line 135.

  1. 139-140, "few of these studies are not population based"... please check this sentence; I suspect you are trying to say "few of these studies are population based..."

Exactly! Thank you for pointing this out. It is corrected now (line 145)

  1. line 162-163  by [an] active search

Done. Thank you (line 170)

  1. I want to make sure I understand line 200-201. You are saying if someone had no symptoms but tested postive they were excluded? Why is this? Is this a World Health Organisation recommendation? Aren't many covid cases asymptomatic? Conversely, why did you include symptomatic negative cases but not asymptomatic postive cases? This could be inflating your death rate of 3.7% could it not? Do you know how many tested positive but were asymptomatic? If so, what happens to the analyses if you include those people? Please help the reader understand this better.

We appreciate your comment, however we considered that given the fact that we were not making universal testing in our patients with IBD, data with regard asymptomatic patients will be completely biased. Therefore, we do not have data on positive asymptomatic cases, as they were excluded from our research. We decided to focus only on symptomatic patients, since they were the challenging patients during the first wave of the pandemic. We have added the following comment on the Discussion section (page 20, lines 20-23): “We decided to exclude asymptomatic patients, as there was no policy for a universal testing in our patients nor in the general population during the first wave. Therefore data on asymptomatic patients with IBD and positive SARS-CoV-2 test is scarce and can imply a selection bias”.

  1. At line 246, what do you mean that there are "60,512 patients actively controlled"? Does that mean each of the 60,512 has a match from the general population?

No, these numbers are in reference to the patients with IBD that are actively controlled and registered in the ENEIDA registry. This is explained in page 6, line 258-259. To clarify this point the word “controlled” has been replaced by “followed-up) (line 263).

  1. SECURE-IBD, whilst mentioned in the introduction, somewhat appears out of nowhere in Table 5. You may want to mention in your methods that you do some comparing to SECURE-IBD and describe it briefly (i.e. a database of people with IBD and Covid-19).

Thank you for your suggestion. We have included this in the Methods section (page 6, line 224-226): “ Data on outcomes of our study were provided to those registered on the SECURE-IBD registry (accessed on 24th of August, 2021)[(9)], considered the worldwide IBD registry on COVID-19”.

  1. You do acknowledge some of your sampling bias issues at line 142. I have noted I wondered why you excluded those who tested positive but had no symptoms at line 200 methods. You also mention only 18% of centres performed an active search of cases. Did you think to compare those who were found via an active search and those who were found via the other means mentioned at line 143, as from that you will be able to see if data collection methods skewed any of your results somehow. e.g., maybe those actively found were less severe.

Thanks for your appreciation. We consider that doing an active search is more likely to find milder patients, who, given the few symptoms in a pandemic situation, have not sought medical attention.  Nevertheless, we did a comparison in outcomes between centers that performed an active search of patients with those who did not do this approach. We did not find differences in the assesses outcomes, therefore we decided not to include this information, to avoid excessive material.

However, if you consider so, we can provide this information as a supplementary table.

Reviewer 2 Report

Dear Editor, 

  This is largest cohort of patients with IBD and COVID-19. The study provide a national wide evidence regarding this important issue. The followings are my comments. 

#1. Tittle of Table 1 should notify IBD patients with COVID infection. i.e. regarding COVID-19 and inflammatory bowel disease to make it more clear. 

#2. It is interesting to know in Table 6 regarding patients requiring Immunosuppression initiation or modification. What changes happened regarding the medication after COVID infection ?More likely to received Vedolizumab/Ustekinumab rather than anti TNF ? in your cohort ?

Author Response

Reviewer 2.

 Dear Editor, 

  This is largest cohort of patients with IBD and COVID-19. The study provide a national wide evidence regarding this important issue. The followings are my comments. 

  1. Tittle of Table 1 should notify IBD patients with COVID infection. i.e. regarding COVID-19 and inflammatory bowel disease to make it more clear.

Thank you for your suggestion. Is it changed as suggested. 

  1. . It is interesting to know in Table 6 regarding patients requiring Immunosuppression initiation or modification. What changes happened regarding the medication after COVID infection ?More likely to received Vedolizumab/Ustekinumab rather than anti TNF ? in your cohort ?

Data is not shown, due the huge amount of data that we are already presenting, but we do not find differences in prescription before and after COVID-19 infection. In Table 6, you can appreciate that the relative proportion of patients under anti-TNF, for example, is the most frequent biological prescription even after COVID-19.

Round 2

Reviewer 1 Report

Thank you for allowing me to see your revision. You have in most cases responded adequately but I do have several follow up queries. This is ensure it is fit for publication and to minimise criticism after it is published.

You numbered your responses so I will number them and respond.

  1. Upon reflection I see the journal does request no headings in the abstract, which is perhaps uncommon, but thank you for what you did do.
  2. I did not realise your abstract was at its limit. I do think mentioning your comparison to the general population is nonetheless warranted given that you have that as a conclusion. You do not mention the comparison to the general population in the results of the abstract yet you mention this in your conclusion. Perhaps you could consider deleting the sentences: "Immunosupression was transiently discontinued..." AND "Long-term sequelae were..." and replace them with a sentence about the comparison to the general population you mention lines 308-313. This will make the conclusion align more with the results. Sorry I did not make this clear in the first iteration but I did not quite spot that then.
  3. When you say reference 49, I think you mean reference 46. Reference 46. That paper looked at hospitalised Covid 19 patients and even acknowledges on page 487 that hospital admission criteria can alter the CFR. If you look at Table 5 of your paper, many of your patients were by definition not hospitalised. Hence your manuscript is looking at all patients with IBD who got Covid 19 whereas reference 46 is looking at all patients in the general population hospitalised for Covid. Obviously not everybody with Covid is hospitalised. I think to compare your study with that paper you need to compare what percentage of your hospitalised patients died to the percentage who died in reference 46. I don't think you can say: "These figures are similar to those...in Spain [(46)]" unless you can point to me where in reference 46 it states what percentage were hospitalised. In general, you need to rethink that whole comparison and ensure you are comparing like with like.
  4. Thank you for clarifying.
  5. Thank you for amending.
  6. Thank you for amending.
  7. Thank you for amending.
  8. Thank you for responding and helping me understand this better. I think adding what you did to the discussion does help the reader understand why you excluded asymptomatic cases, even if they happen to disagree with that reasoning. You may want to consider having this in your limitations paragraph as opposed to your opening paragraph of discussion but I will leave that up to you.
  9. Thank you for amending this.
  10. I have just one thing to clarify. You say you "provided to... the SECURE-IBD registry".  Do you mean you compared your results to SECURE-IBD? If so, consider changing the key word to compare in the main text and title for Table 5. It appears you are comparing them in Table 5.
  11. I understand your concerns regarding having too many results. I think it would be worthwhile nonetheless to add a blanket sentence basically saying you compared centres that actively searched for cases to those that did not and found no differences in outcome A, outcome B and outcome C (I assume these are death, hospitalisation, ICU admission and assume all p-values are > 0.05). It will only take up 10-30 words but is valuable information; a full table or even supplementary material would not be required.

Author Response

Dear reviewer 1

Thank you so much for your critical revision. As state previously we really think that your comments and suggestions have improve the quality of our manuscript.

We have addressed your comments as follows:

Thank you for allowing me to see your revision. You have in most cases responded adequately but I do have several follow up queries. This is ensure it is fit for publication and to minimise criticism after it is published.

You numbered your responses so I will number them and respond.

  1. Upon reflection I see the journal does request no headings in the abstract, which is perhaps uncommon, but thank you for what you did do.

No problem, thank you for pointing it out

  1. I did not realise your abstract was at its limit. I do think mentioning your comparison to the general population is nonetheless warranted given that you have that as a conclusion. You do not mention the comparison to the general population in the results of the abstract yet you mention this in your conclusion. Perhaps you could consider deleting the sentences: "Immunosupression was transiently discontinued..." AND "Long-term sequelae were..." and replace them with a sentence about the comparison to the general population you mention lines 308-313. This will make the conclusion align more with the results. Sorry I did not make this clear in the first iteration but I did not quite spot that then.

As suggested, we removed the phrase on immunosuppression and long-term sequelae and state the differences in age between patients with IBD and the general population. We agree with the reviewer that this is more in line with the conclusions.

  1. When you say reference 49, I think you mean reference 46. Reference 46. That paper looked at hospitalised Covid 19 patients and even acknowledges on page 487 that hospital admission criteria can alter the CFR. If you look at Table 5 of your paper, many of your patients were by definition not hospitalised. Hence your manuscript is looking at all patients with IBD who got Covid 19 whereas reference 46 is looking at all patients in the general population hospitalised for Covid. Obviously not everybody with Covid is hospitalised. I think to compare your study with that paper you need to compare what percentage of your hospitalised patients died to the percentage who died in reference 46. I don't think you can say: "These figures are similar to those...in Spain [(46)]" unless you can point to me where in reference 46 it states what percentage were hospitalised. In general, you need to rethink that whole comparison and ensure you are comparing like with like.

You are completely right: it is reference 46 and we apologize for this confusion. However, we agree with the reviewer that this is not the best reference for demonstrating that our numbers are like those found in Spain on the first wave. Therefore we decided to change this reference (Casas-Rojo JM, Antón-Santos JM, Millán-Núñez-Cortés J, Lumbreras-Bermejo C, Ramos-Rincón JM, Roy-Vallejo E, et al. Clinical characteristics of patients hospitalized with COVID-19 in Spain: Results  from the SEMI-COVID-19 Registry. Rev Clin Esp. 2020 Nov;220(8):480–94) for this: Lia Alves-Cabratosa, Marc Comas-Cufí, Jordi Blanch, et al. Persons with SARS-CoV-2 during the First and Second Waves in Catalonia (Spain): A Retrospective Observational Study Using Daily Updated Data, JMIR Public Health Surveill. 2021 Nov 16. doi: 10.2196/30006. Online ahead of print.

As it is more in line with our own population. In this last article, that compare outcomes from the first and second wave, all outcomes are analysed in confirmed cases (most of our cases are confirmed COVID-19 cases). On the first wave they shown that 19.5% of the general population required hospitalisation, 2.1% ICU and 4.6% die, numbers very similar to ours. We are aware that this article studied only population of Girona (Catalonia) but we consider it as good representation of the Spanish population as Catalonia is one of the most affected location by the pandemic. We have, therefore, include this reference and data in the abstract, the methods section (line 226-228) as well as results (lines 389-392)

  1. Thank you for clarifying.
  2. Thank you for amending.
  3. Thank you for amending.
  4. Thank you for amending.
  5. Thank you for responding and helping me understand this better. I think adding what you did to the discussion does help the reader understand why you excluded asymptomatic cases, even if they happen to disagree with that reasoning. You may want to consider having this in your limitations paragraph as opposed to your opening paragraph of discussion but I will leave that up to you.

Thanks for your suggestion but, if you do not mind, we prefer to leave it as is.

  1. Thank you for amending this.
  2. I have just one thing to clarify. You say you "provided to... the SECURE-IBD registry".  Do you mean you compared your results to SECURE-IBD? If so, consider changing the key word to compare in the main text and title for Table 5. It appears you are comparing them in Table 5.

We have changed to COMPARE, both in the text (line 226) as well as in Table 5, as suggested with the reviewer

  1. I understand your concerns regarding having too many results. I think it would be worthwhile nonetheless to add a blanket sentence basically saying you compared centres that actively searched for cases to those that did not and found no differences in outcome A, outcome B and outcome C (I assume these are death, hospitalisation, ICU admission and assume all p-values are > 0.05). It will only take up 10-30 words but is valuable information; a full table or even supplementary material would not be required.

We are very thankful for this last comment. We have stated this issue as suggested as follows (lines 400-403): There were no difference on the proportion of hospitalisation, ICU admission, severe COVID-19 or death between patients found under active search (18% of the centres) vs. those that were found by direct notification from the patient itself, the family physician, the emergency department or the hospitalisation unit (data not shown).

In lines 122 and 123 we also state that fact on the limitations of our study
